# Structure Activity Relationship of Key Heterocyclic Anti-Angiogenic Leads of Promising Potential in the Fight against Cancer

**DOI:** 10.3390/molecules26030553

**Published:** 2021-01-21

**Authors:** Hossam Nada, Ahmed Elkamhawy, Kyeong Lee

**Affiliations:** 1College of Pharmacy, Dongguk University-Seoul, Goyang 10326, Korea; hossam_hammouda@dgu.ac.kr (H.N.); a_elkamhawy@mans.edu.eg (A.E.); 2Department of Pharmaceutical Organic Chemistry, Faculty of Pharmacy, Mansoura University, Mansoura 35516, Egypt

**Keywords:** anticancer, heterocyclic, anti-angiogenics, structure–activity relationship, in silico pharmacokinetics, molecular modelling

## Abstract

Pathological angiogenesis is a hallmark of cancer; accordingly, a number of anticancer FDA-approved drugs act by inhibiting angiogenesis via different mechanisms. However, the development process of the most potent anti-angiogenics has met various hurdles including redundancy, multiplicity, and development of compensatory mechanisms by which blood vessels are remodeled. Moreover, identification of broad-spectrum anti-angiogenesis targets is proved to be required to enhance the efficacy of the anti-angiogenesis drugs. In this perspective, a proper understanding of the structure activity relationship (SAR) of the recent anti-angiogenics is required. Various anti-angiogenic classes have been developed over the years; among them, the heterocyclic organic compounds come to the fore as the most promising, with several drugs approved by the FDA. In this review, we discuss the structure–activity relationship of some promising potent heterocyclic anti-angiogenic leads. For each lead, a molecular modelling was also carried out in order to correlate its SAR and specificity to the active site. Furthermore, an in silico pharmacokinetics study for some representative leads was presented. Summarizing, new insights for further improvement for each lead have been reviewed.

## 1. Introduction

As the second leading cause of mortality globally, cancer has become the focus for extensive research [1,2]. Although cancer progression and metastasis consist of multiple, complex, interacting, and interdependent steps [3], angiogenesis plays an essential part of the tumor’s growth and metastasis [4], owing to the fact that growth beyond the size of l–2 mm^3^ requires tumors to develop an adequate blood supply [5,6]. Angiogenesis is the processes whereby new blood and lymphatic vessels form [7]. Angiogenesis and its induction remain a major hallmark of cancer as it flourishes nutrient-deprived tumors with oxygen and nutrients, thus routing tumor metastasis [8,9].

Under normal conditions, angiogenesis is essential for formation of a new vascular network to supply nutrients, oxygen, and immune cells, as well as to remove waste products [10]. This process is regulated by a balance between pro- and anti-angiogenic molecules, and once that delicate balance is disturbed [11], it could lead to various diseases, especially cancer [12]. Angiogenesis is a vital mediator of tumor development [13]. As tumors enlarge, diffusion distances from the current vascular supply rise, leading to hypoxia [14]. Continued expansion of a tumor mass needs new blood vessel formation to offer rapidly proliferating tumor cells with a suitable supply of oxygen and metabolites [15]. Without the proper blood supply, tumors cannot grow beyond a critical size or metastasize to another organ [16]. Therefore, targeting angiogenesis is one of the most effective ways to stop a tumor progression [17].

Most of the drugs that have been developed to combat angiogenesis have heterocyclics in their backbone as a common feature. This is demonstrated by heterocyclics occupying the major part of the number of FDA-approved anti-angiogenic drugs, as demonstrated in Figure 1. This domination of FDA-approved drugs is likely due to the wide availability of various heterocyclic fragments that have different potency, physicochemical properties, and lipophilicity. This versatility makes heterocyclic fragments a sound rational choice when developing new drugs or altering an existing one.

In order to develop more potent and targeted anti-angiogenics, a proper understanding of their structure–activity relationship is required. In this review, we discuss some of the most potent heterocyclic anti-angiogenics. In each class of heterocyclics, we illustrate the structure–activity relationship (SAR) of some promising leads that have been chosen thanks to their future potential for development into more potent anti-angiogenics as well as the availability of enough supporting data to predict their SAR. A molecular modeling was then conducted to illuminate how these leads interact with their active sites and which parts should be improved in order to obtain a more potent and specific drug candidate.

## 2. Nitrogen-Based Heterocycles

The number of anti-cancer candidates possessing a nitrogen heterocycle is an indicator of the structural significance of nitrogen-based heterocycles in the fight against cancer [20,21,22]. More than 75% of drugs approved by the FDA and currently available in the market are nitrogen-containing heterocyclic moieties [23]. One of the most important targets in angiogenesis inhibition is vascular endothelial growth factor receptors (VEGFRs).

Vascular endothelial growth factor is an important signaling protein involved in both vasculogenesis (the formation of the circulatory system) and angiogenesis (the growth of blood vessels from pre-existing vasculature) [24,25]. As its name implies, VEGF activity is restricted mainly to cells of the vascular endothelium [26]. The expression of VEGF is potentiated in response to hypoxia, by activated oncogenes, and by a variety of cytokines [27]. VEGF induces endothelial cell proliferation, promotes cell migration, and inhibits apoptosis [28]. VEGF induces angiogenesis as well as permeabilization of blood vessels, and plays a central role in the regulation of vasculogenesis [29,30].

Deregulated VEGF expression contributes to the development of solid tumors by promoting tumor angiogenesis and to the etiology of several additional diseases characterized by abnormal angiogenesis [31]. Consequently, inhibition of VEGF signaling abrogates the development of a wide variety of tumors [32]. All members of the VEGF family stimulate cellular responses by binding to tyrosine kinase receptors (the VEGFRs) on the cell surface, causing them to dimerize and become activated through transphosphorylation [33,34].

Several nitrogen-based heterocyclic drugs have been used to inhibit VEGF, for example, agents that inhibit the VEGFR tyrosine kinase such as the pyrrolidinone-based Sunitinib were approved by the FDA for the treatment of renal cell carcinoma (RCC) and imatinib-resistant gastrointestinal stromal tumor [35,36]. Another example is the phthalazine-based vatalanib, which is under investigation for the treatment of metastatic colorectal cancer and non-small cell lung cancer (NSCLC) [37,38]. Indole ring is one of the nitrogen-based heterocyclics that have been involved in VEGF inhibition [39]. It has demonstrated the ability to inhibit the proliferation, growth, and invasion of human cancer cells. Panobinostat is an indole-based drug approved in 2015 for the treatment of multiple myeloma [40].

Renhowe et al. synthesized a series of quinolin-2-one analogues that showed promising VEGFR-2 inhibition activity (Table 1) [41]. Upon examining the series of analogues synthesized by Renhowe et al., it was found that the free NH of the hydroquinolin-2-one scaffold (Figure 2), the quinolinone carbonyl, and the benzimidazole NH formed donor–acceptor motifs that would bind to the hinge region of the VEGFR-2, which could be implicated in tumor vasculature formation and maintenance [42,43]. The importance of a hydrogen donating group was found through substituting different groups at the C4 position, observing the potency increasing with (NH_2_ > OH > H). The incorporation of large basic amine at the C4 position such as an amino quinuclidine resulted in much more potent activity; however, it negatively affected the pharmacokinetics. Therefore, the NH_2_ substitution at C4 was maintained and subsequent efforts focused on modifying the substitution on both rings A and D to attain additional enzymatic affinity and to improve physicochemical properties.

As illustrated in Figure 3, the interaction of compound **10** with VEGFR-2 receptor active site (IC_50_ = 0.026 μM) was subjected to docking studies using Schrodinger maestro. The modelling simulation shows a significant docking score of −9.058, along with the formation of two hydrogens bonds as well as two π–cation bonds. These interactions were found to be energetically significant as all the formed bonds among the donor and acceptor atoms are within about 3.7 Å.

Although VEGF is an important pathway in angiogenesis, several other targets are also involved in angiogenesis. Matrix metalloproteinases (MMPs) are one such target involved in angiogenesis [44]. MMPs are implicated in early steps of tumor evolution including stimulation of cell proliferation and modulation of angiogenesis [45]. During angiogenesis, new vessels develop from present endothelial lined vessels to encourage the degradation of the vascular basement membrane and remodel the extracellular matrix (ECM), followed by endothelial cell migration, proliferation, and formation of the new generation of matrix components [46,47].

Matrix metalloproteinases participate in the disruption, tumor neovascularization, and successive metastasis, while tissue inhibitors of metalloproteinases (TIMPs) downregulate the activity of these MMPs [48]. Therefore, MMPs, through the modulation of the balance between pro- and anti-angiogenic factors, can directly or indirectly mediate the angiogenic response [49].

The relation between MMP overexpression in tumor and cancer progression has encouraged the progress of preclinical trials with a series of inhibitors designed to block the proteolytic activity of these enzymes.

Thus far, no FDA-approved MMP inhibitors for the treatment of cancer have emerged. However, several leads have been developed; for example, Becker et al. reported new piperidine α-sulfone hydroxamates with potent matrix metalloproteinase inhibition activity [50]. The α- and β-sulfone derivatives showed a difference in the inhibitory activity toward the metalloproteinases. The β-sulfone hydroxamate showed potency for the targeted MMPs and selectivity for MMP-1, but generally exhibited poor oral bioavailability. In addition, some β-sulfones with α-hydrogens can undergo β-elimination. On the other hand, the α-sulfones possess both potency and selectivity and provide an improvement in oral exposure demonstrated by a higher C_max_ value and bioavailability relative to the β-sulfones.

On further investigation, they concluded that α-sulfone derivatives are two to four times more potent and bioavailable than the β-sulfones. The higher bioavailability and C_max_ may be due to greater steric bulk around the hydroxamate, protecting it from the usual modes of hydroxamate metabolism including N-O bond cleavage, hydrolysis, and glucuronidation. The findings of Becker et al. can be explained through the comparison of the different activities of α- and β-sulfone derivatives, as demonstrated in Table 2.

Looking at the interaction of the α- and β-sulfone derivatives (Figure 4 and Figure 5) with the active site residues of MMP-2 [51], the difference in potency can be elucidated because of the fact that the α-derivative showed a docking score of −7.054, while its counterpart, the β-derivative, exhibited a docking score of −5.22. This difference in the docking scores as well as forming different interactions with the active site could explain why the α-sulfone derivative would be a more promising lead for further development.

## 3. Heterocyclic Sulfonamides

Sulfonamides significance stems from the fact that they constitute a significant class of drugs with various types of pharmacological effects including antitumor, anti-carbonic anhydrase, and diuretic activity [53,54,55,56]. There are many reports of a multitude of structurally novel heterocyclic sulfonamide derivatives that have been stated to display significant antitumor activity [57,58]. Although the mechanism by which they combat cancer may vary, the majority of anti-cancer sulfonamides act on angiogenesis through several pathways.

As of today, there are no FDA-approved heterocyclic sulfonamide anti-angiogenics in the market, but there are several promising leads being developed. One of such leads is E7820, which is now in phase II clinical trial for colorectal cancer treatment through the inhibition of integrin α2 [59]. Integrins are transmembrane receptors that are central to the biology of many human pathologies, where they act through the mediation of the cell-extracellular matrix and cell–cell interaction [60]. This mediative action is carried out through the transduction of information from the extracellular environment to modulate cell responses, together with adhesion, spreading, migration, growth signaling, survival signaling, secretion of proteases, and invasion [3,61,62]. Numerous studies report that the increased levels of integrin α2 facilitate the spread of cancer [63,64,65,66]. Thus, inhibitors of integrin α2, such as that shown in Figure 6, will slow the spread of cancer and its metastasis.

Hypoxia inducible factors (HIFs) are another promising anti-angiogenic inhibitor [67]. This is because of the requirement of an adequate oxygen supply for the macroscopic tumor to grow [68]. This need is fulfilled through tumor angiogenesis, which results from an increased synthesis of angiogenic factors and a decreased synthesis of anti-angiogenic factors [49,69].

The shift of the balance between pro- and anti-apoptotic factors due to the metabolic adaptation of tumor cells to decrease the oxygen availability by increasing glucose transport and glycolysis to promote survival is prominent in hypoxia [69,70]. In this regard, HIF-1, which is induced by many factors, is mainly implicated in tumor angiogenesis.

A class of heterocyclics based on the lead sulfonamide hits identified by Gerguson et al. showed potent activity against hypoxia (Table 3) [71,72]. It was found that the oxygen atom in ring A was essential for the activity, while the double bond in the same ring resulted in increased hepatotoxicity, leading to the conclusion that the double bond should be removed. Furthermore, the sulfonamide moiety as well as the hydrophobic substitutions in ring B increased the activity significantly. In ring C, the methoxy substituent at *para* position is essential for activity, while *meta* methoxy is not. Additionally, upon replacing the *para* methoxy with electron withdrawing groups, the activity was lost, demonstrating that only electron donating groups should occupy this position. This SAR is summarized in Figure 7.

The interaction of compound **13**, possessing IC_50_ of 0.6 μM, with HIF-1 active site residue, is demonstrated in Figure 8. It showed better binding energy and inhibition constant as compared with N-((2,2-dimethylchroman-6-yl) methyl) sulfonic amide, with docking scores of −6.85 and −5.4, respectively. This could be owing to the ability of the sulfonamide derivative (**13**) to form a hydrogen bond and metal interaction with the active site residue.

## 4. Oxygen-Based Heterocycles

About 15% of all FDA-approved drugs are oxygen-based heterocycles [73]. To date, there are several FDA-approved oxygen-based anti-angiogenic cancer drugs. Some of these drugs (Figure 9) are based on the oxetane ring such as paclitaxel (PTX, Taxol^®^) and cabazitaxel (Jevtana) acting via microtubule-targeting [74].

Microtubules are cytoskeletal elements that are necessary for many functions including intracellular transport, motility, morphogenesis, and cell division [75,76]. α–β tubulin heterodimers make up the microtubules by assembling in sequence to form the protofilaments of the tube [77]. This microtubule polymerization leads to a characteristic heterogeneity among the two ends of the microtubule, giving rise to different kinetics of the addition and subtraction of heterodimers at the two ends [78]. At the ‘plus’ end, the kinetics of the polymerization and depolymerization are faster than those at the other end, the slower so-called ‘minus’ end. Inside the cell, microtubules are attached by their minus ends at the microtubule-organizing center, placing their plus ends to the cell border [79]. The plus ends constantly grow and shorten, which is a property vital for various cellular processes including cell division [80].

Throughout mitosis, normal cells could arrest owing to interference in the dynamic properties of microtubules, where microtubules continuously polymerize and rapidly depolymerize, making a ‘pulling device’ for the duplicated chromosomes [81]. MTAs (microtubule-targeting agents) act through the stabilization of microtubules against depolymerization [82,83,84]. One of the most promising microtubule targeting agents is paclitaxel. In a study conducted by Ganesh et al., the cytotoxicity of some paclitaxel derivatives was measured as illustrated in Table 4 [85].

From the study of Ganesh et al., it was found that any changes to rings C and D led to loss of activity, indicating the importance of their structural rigidity. The following groups were found to be essential for activity: the N-acyl group, the free 2′-hydroxy group, and the acyloxy group substitution on ring E. In addition, the *ortho* and *meta* substitutions on ring E increased the activity. On the other hand, when the acetate group on ring D was removed, the activity decreased significantly. The SAR of the above-mentioned paclitaxel analogues is illustrated in Figure 10.

The interaction of paclitaxel with microtubule active site (Figure 11) further illuminated this SAR, where paclitaxel showed a very high binding affinity with the active site (docking score of −9.48) [87]. Furthermore, paclitaxel formed three hydrogen bonds and one π–π interaction within the active site, showing a remarkable ability to interact with the microtubule active site.

## 5. Drug Likeness and Absorption Distribution Metabolism Excretion (ADME) Prediction of the Chosen Inhibitors

Computational approaches have become an essential part of interdisciplinary drug discovery research. Understanding the science behind computational tools and their opportunities is essential to make a real outcome on drug discovery at different stages. If applied in a scientifically meaningful way, computational methods improve the ability to identify and evaluate potential drug molecules [90,91,92,93,94,95]. A good antagonistic interaction of inhibitors with a receptor protein or enzyme does not assure the capability of an inhibitor as a drug; consequently, absorption distribution metabolism excretion (ADME) analysis is important in the drug development [96]. ADME is based on Lipinski’s rule of five and assists in the approval of inhibitors for biological systems.

One of the major causes of the failure of most medicines in clinical experiments is having poor ADME characteristics and unfavorable toxicology [97]. Apart from efficacy and toxicity, various drug development failures are due to poor pharmacokinetics and bioavailability [98]. Gastrointestinal absorption and brain access are two pharmacokinetic behaviors crucial to be evaluated at various stages of the drug discovery [99]. Therefore, four leads belonging to each class were subjected to an in silico pharmacokinetic study using the SwissADME server (compound **10** as representative for nitrogen-based drugs (referred as molecule 1 in the BOILED-Egg chart), compound **12** as a representative for MMPIs (molecule 2), compound **13** as a representative for the heterocyclic sulfonamides (molecule 3), and paclitaxel as a representative for oxygen-based heterocycles (molecule 4)). The pharmacokinetic study is demonstrated in Figure 12 using the brain or intestinal estimated permeation method (BOILED-Egg).

Upon looking at the BOILED-Egg chart, we can see that, even though the four leads exert their pharmacological action through different mechanisms, three compounds (**10**, **12**, and **13**) were found to obey Lipinski’s rule of five, indicating their absorbance in the GIT if taken orally, but avoiding any side effects resulting from passing the blood–brain barrier (BBB). However, compound **10** is predicted to be expelled from the central nervous system by P-glycoprotein (Pgp), resulting in a possible poor bioavailability. Even though paclitaxel showed promising biological activity, it fails to obey Lipinski’s rule of five, leading to poor oral bioavailability.

## 6. Conclusions

Throughout this review, several leads were investigated for SAR, molecular interaction, and in silico pharmacokinetics study. The following outlines were concluded, aiming to develop more active analogues of the discussed leads. Compound **10** and its derivatives would benefit from both a modification to prevent its efflux by the PGP as well as modifications to increase its interaction with the VEGFR-2 active site. Both compounds **12** and **13** and their derivatives show good drug-likeness properties, but would benefit from further isosteric changes to their structures to increase their interactions with their respective active site. Paclitaxel and its derivatives have shown the highest binding potential to MTA active site residue, with docking scores around −9.5. However, they all violated Lipinski’s rule of five, leading to poor bioavailability. Therefore, further modification to their chemical structure is highly needed.

## Figures and Tables

**Figure 1 molecules-26-00553-f001:**
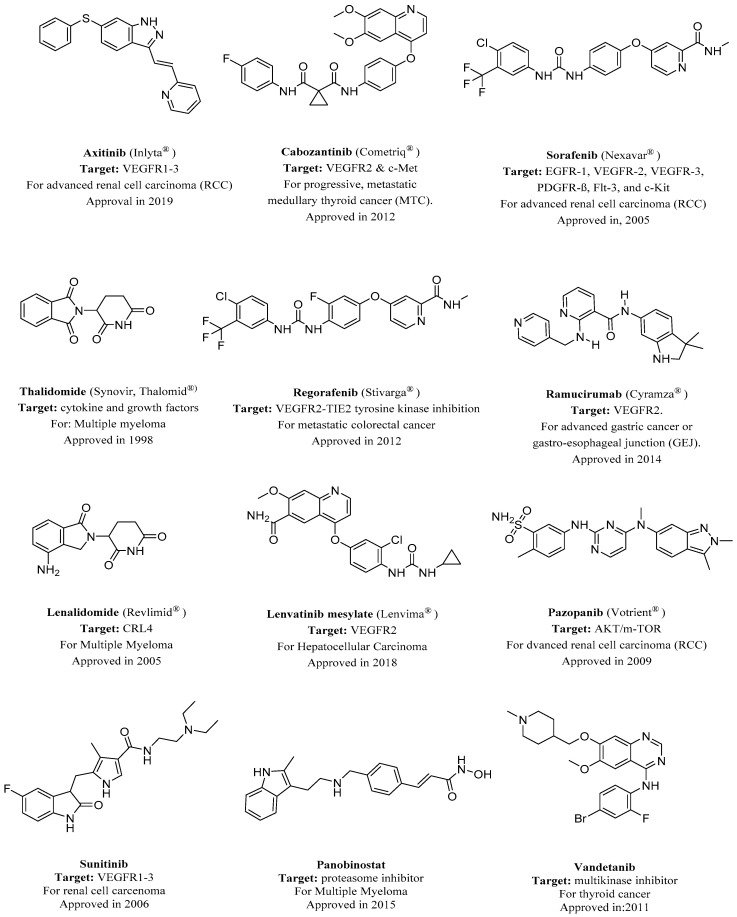
FDA-approved angiogenesis inhibitors for treatment of cancer [18,19]. VEGFR, vascular endothelial growth factor receptor.

**Figure 2 molecules-26-00553-f002:**
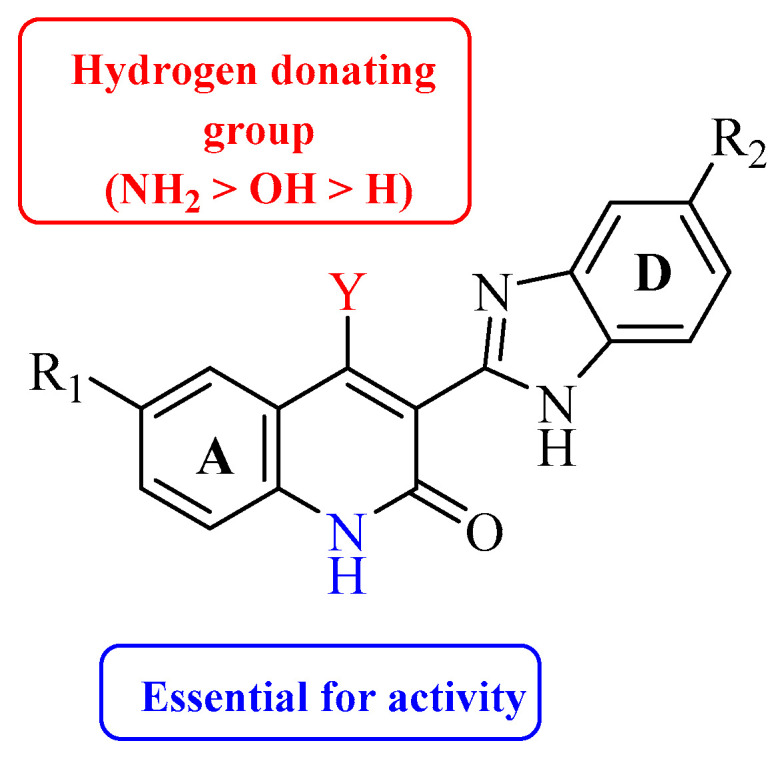
Structure–activity relationship study of 3-benzimidazol-2- ylhydroquinolin-2-one derivatives.

**Figure 3 molecules-26-00553-f003:**
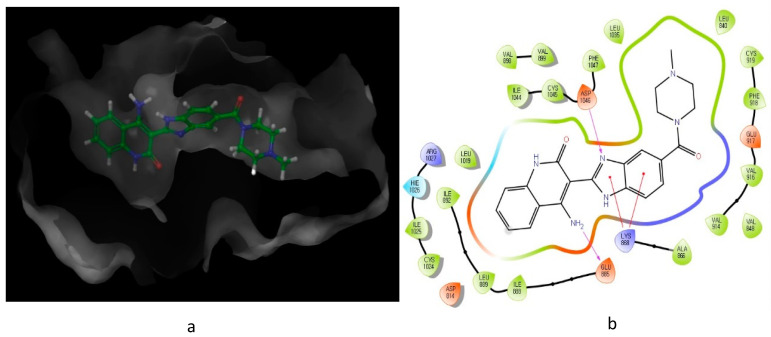
Predicted interaction of compound **10** (imidazole-scaffold) with active site residues of VEGFR-2. (**a**) 3D structural view of ligand inside the receptor’s active site. (**b**) Ligand interaction diagram; violet lines represent hydrogen bonds and red lines represent π–cation interactions.

**Figure 4 molecules-26-00553-f004:**
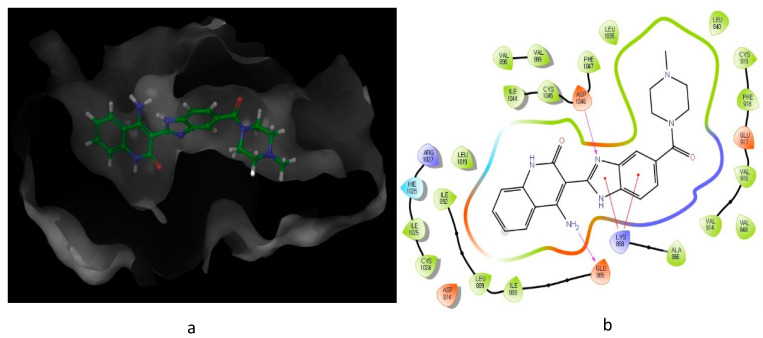
Predicted interaction of α-sulfone (**12**) derivative active site residues of matrix metalloproteinase (MMP) [51,52]. (**a**) 3D structural view of ligand inside the receptor’s active site. (**b**) Ligand interaction diagram; violet lines represent hydrogen bonds and green lines and grey line represent metal interactions.

**Figure 5 molecules-26-00553-f005:**
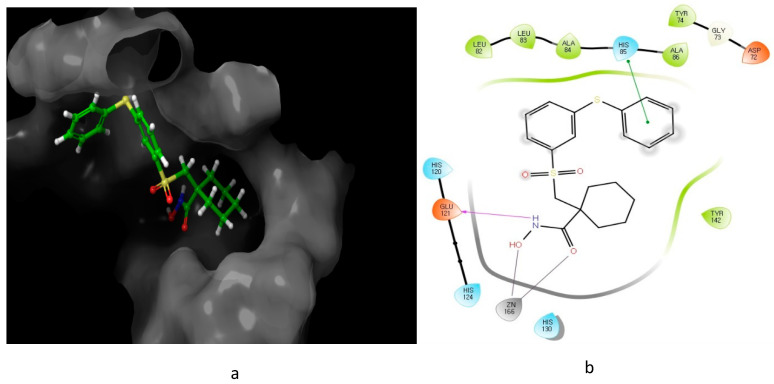
Predicted interaction of β-sulfone (**11**) derivative active site residues of MMP. (**a**) 3D structural view of ligand inside the receptor’s active site. (**b**) Ligand interaction diagram; violet lines represent hydrogen bonds, green lines represent π–π interactions, and grey line represent metal interactions.

**Figure 6 molecules-26-00553-f006:**
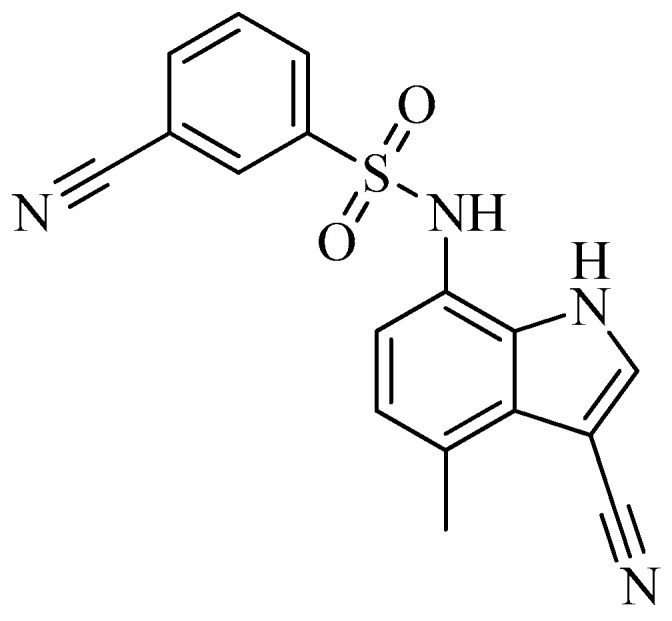
Chemical structure of the heterocyclic sulfonamide anti-angiogenic E7820.

**Figure 7 molecules-26-00553-f007:**
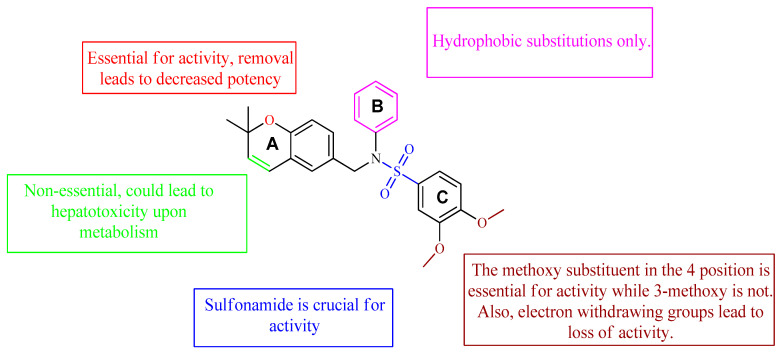
Structure–activity relationship study of the benzenesulfonamide hits and its analogues [71,72].

**Figure 8 molecules-26-00553-f008:**
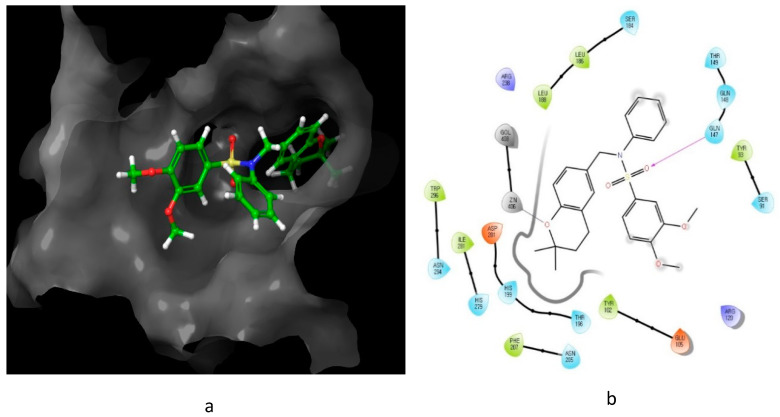
Predicted interaction of compound **13** with hypoxia-inducible factor active site residue. (**a**) 3D structural view of the sulfonamide ligand inside the receptor’s active site. (**b**) Ligand interaction diagram; violet lines represent hydrogen bonds and grey lines represent metal interaction.

**Figure 9 molecules-26-00553-f009:**
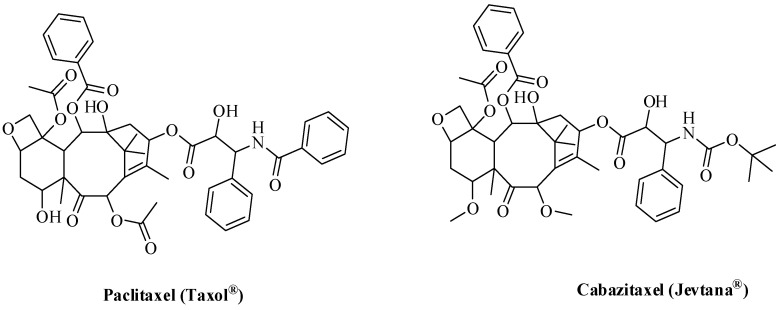
FDA-approved oxygen-based anti-angiogenics.

**Figure 10 molecules-26-00553-f010:**
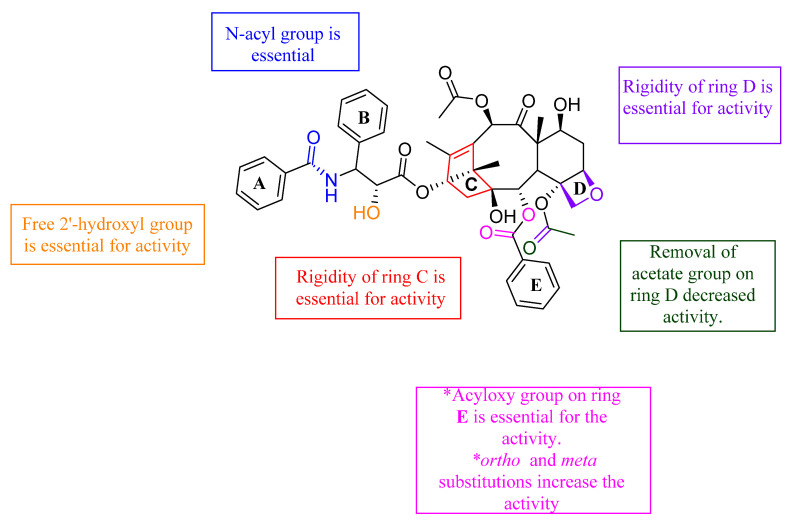
Structure–activity study of paclitaxel analogues [85,86].

**Figure 11 molecules-26-00553-f011:**
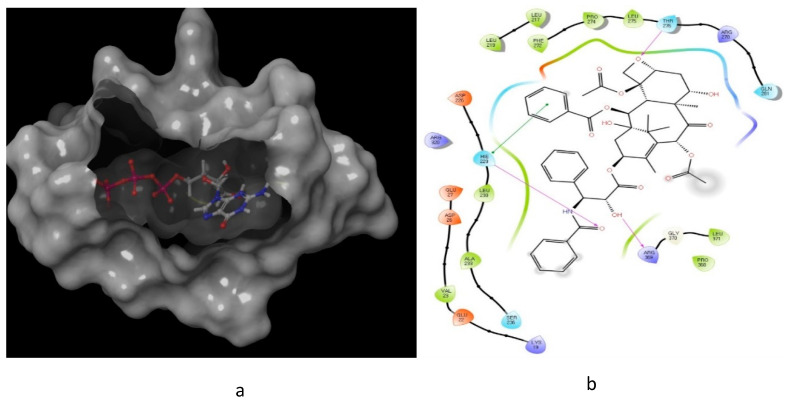
Predicted interaction of paclitaxel with microtubule active site residue [88,89]. (**a**) 3D structural view of the sulfonamide ligand inside the receptor’s active site. (**b**) Ligand interaction diagram; violet lines represent hydrogen bonds and green lines represent π–π interactions.

**Figure 12 molecules-26-00553-f012:**
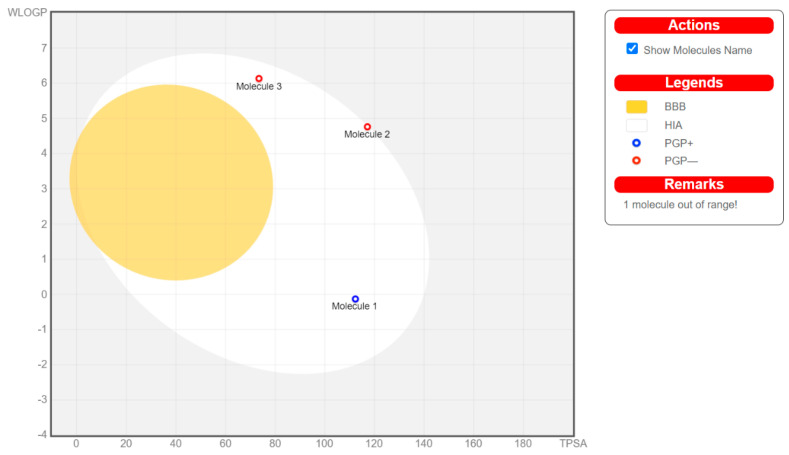
Brain or intestinal estimated permeation method (BOILED-Egg) chart for compounds **10**, **12**, **13**, and paclitaxel.

**Table 1 molecules-26-00553-t001:** In vitro inhibition of vascular endothelial growth factor receptor 2 (VEGFR-2) by the quinolin-2-one analogues (**1**–**10**) [41].

Compound	Y	R_1_	R_2_	VEGFR-2 (IC_50_ μM)	PDGFRβ (IC_50_ μM)
**1**	OH	H	H	0.24	0.020
**2**	NH_2_	H	H	0.058	0.010
**3**	NHMe	H	H	0.22	0.030
**4**	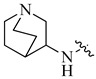	H	H	0.17	0.0002
**5**	H	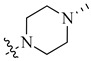	H	0.017	0.0003
**6**	NH_2_	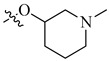	H	0.005	0.0001
**7**	NH_2_	H	Me	0.22	0.030
**8**	NH_2_	H	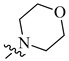	0.057	0.005
**9**	NH_2_	H	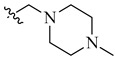	0.027	0.002
**10**	NH_2_	H	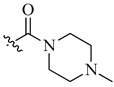	0.026	0.0009

**Table 2 molecules-26-00553-t002:** Comparing α- and β-sulfone derivatives for their matrix metalloproteinase 1 (MMP-1) activity [50].

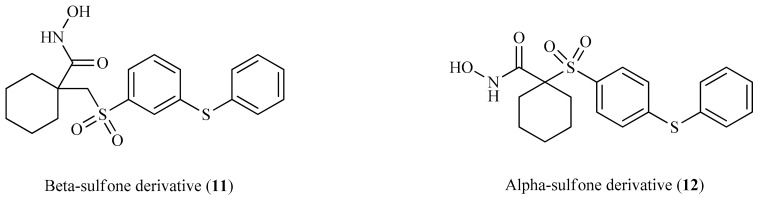						
		**Ki (nM)**				
**Compound**	**MMP-1**	**MMP-2**	**MMP-3**	**MMP-9**	**MMP-13**	**C_max_**
**11** (β-sulfone)	800	0.4	17.5	1	0.6	1372
**12** (α-sulfone)	435	˂0.1	18.1	0.3	0.15	3119

**Table 3 molecules-26-00553-t003:** In vitro inhibition of hypoxia inducible factor 1 (HIF-1) transcriptional activity in cell-based HRE reporter assay by the benzenesulfonamide analogues (**13**–**18**) [72].

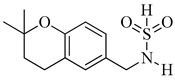
**Compound**	**R_1_**	**R_2_**	**LN229-HRE-Lux (IC_50_ μM)**
**13**	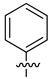	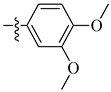	0.6 μM
**14**	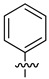	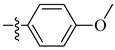	0.6 μM
**15**	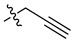	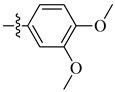	1.3 μM
**16**	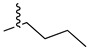	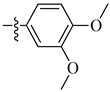	3.3 μM
**17**	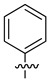	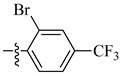	>25 μM
**18**	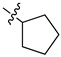	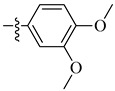	0.5 μM

**Table 4 molecules-26-00553-t004:** Cytotoxic activity of some paclitaxel analogues (**19**–**24**) [85].

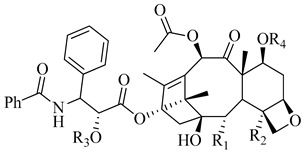
**Compound**	**R_1_**	**R_2_**	**R_3_**	**R_4_**	**IC_50_ PC3 (nM)**
**19**	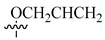	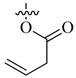	H	H	550
**20**	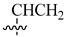	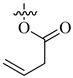	H	H	320
**21**	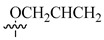	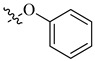	H	H	128
**22**	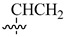	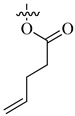	H	H	55
**23**	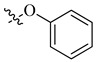	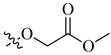	H	H	53
**24**	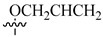	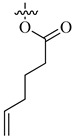	H	H	4600

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
