# Peer review of "Structure Activity Relationship of Key Heterocyclic Anti-Angiogenic Leads of Promising Potential in the Fight against Cancer"

_molecules, 2021, doi:10.3390/molecules26030553_

Round 1

Reviewer 1 Report

The manuscript molecules-1035500 devoted the actual field of the medicinal chemistry, namely current status of heterocyclic anti-angiogenic leads as promising anticancer agents and can be interested to the specialists working in this field. The author’s opinion is clear and based on a wide range of publications. I am personally impressed by the structure of the article, the systematization of scientific data and the sequence of its presentation. The paper fit the Journal scope and formal requirements. However, it needs minor revision before publication.

To improve the quality and perception of the manuscript I would suggest paying attention to following comments:

  1. Figures 1 and 2 should be combined for better perception of the material.
  2. References list should be carefully checked and journal style policy should be strictly followed (all authors, DOI, etc).
  3. There are numerous grammar and orthographical errors in the manuscript (especially references), which should be corrected.

My decision is minor revision.

Author Response

Response to Reviewers' comments

We would like to thank the respected editors and reviewers for their comments on our work.

  • In the manuscript, the corrected sentences have been highlighted in blue color.
  • Improved manuscript editing and updating references has been carried out.
  • In this file, we have replied to all reviewers comments.

Comments of Reviewer 1#

The manuscript molecules-1035500 devoted the actual field of the medicinal chemistry, namely current status of heterocyclic anti-angiogenic leads as promising anticancer agents and can be interested to the specialists working in this field. The author’s opinion is clear and based on a wide range of publications. I am personally impressed by the structure of the article, the systematization of scientific data and the sequence of its presentation. The paper fit the Journal scope and formal requirements.

Reply to reviewer’s comment: We are thankful to the respected reviewer for his opinion on our work, comments and requested corrections.

There are few minor notes:

  1. Figures 1 and 2 should be combined for better perception of the material.

Reply to reviewer’s comment: We have combined both figures as guided by the reviewer.

  1. References list should be carefully checked and journal style policy should be strictly followed (all authors, DOI, etc).

Reply to reviewer’s comment: MDPI reference style format was applied to all references.

  1. There are numerous grammar and orthographical errors in the manuscript (especially references), which should be corrected.

Reply to reviewer’s comment: We thank the respected reviewer for pointing out these errors, and we have updated all the references to MDPI format.

All modifications have been based on the reviewer’s suggestions. In addition, we have checked the whole manuscript.

Finally, we would like to express our thanks to the reviewers and editors for their efforts and comments which helped us to enhance the quality of our manuscript. We hope that our revised manuscript would be satisfactory and would be accepted for publication in Molecules.

Reviewer 2 Report

The manuscript described the recent anti-angiogenics drugs. Angiogenesis is a process essential for cancer growth. The authors showed the structure-activity relationship (SAR) among the existing anti-angiogenics drugs. According to such SAR analysis, a molecular modelling was also carried out. Thus, these findings will be useful for the drug development in anti-angiogenics drugs. Therefore, the manuscript is not too excellent to be published. In other words, the manuscript is so excellent that it should be published.

Comments

(1) It is true that heterocyclic fragments play a crucial role in pharmacodynamic aspects, but they do not always do in pharmacokinetic aspects due to the membrane permeability, solubility, and so on, mainly based on hydrophilicity. Can the calculation in silico pick up pharmaceutical structures sufficient to satisfy both pharmacodynamic and pharmacokinetic aspects?

(2) Which administration routes are suitable for anti-angiogenesis drugs? Is that Intravenous route or/and oral route?

(3) Do intravenously administered angiogenesis inhibitors through VEGFR on the endothelial surface elicit their activity without being transported across the membrane? In that case, a molecular modelling might be conducted just based on pharmacodynamic aspects.

(4) According to the molecular modelling results, do the existing anti-angiogenics drugs share the same binding sites, in each category such as VEGFR inhibitors and MMP inhibitors?

(5) The quinolin-2-one analogues were introduced in this review. They were reviewed in the article titled as “The structural use of carbostyril in physiologically active substances” (https://doi.org/10.1016/j.bmcl.2015.06.027). Why don’t you add this in the reference list?

(6) In line of 21, “the structure activity relationship” is preferable to be “the structure-activity relationship”.

(7) In line of 60, “Molecular modeling” is preferable to be “molecular modeling”.

(8) In lines of 69-70, “VEGFR receptor” is preferable to be “VEGFR”.

(9) In line of 84, “Several Nitrogen based heterocyclic drugs” is preferable to be “Several nitrogen based heterocyclic drugs”.

(10) In line of 86, “Imatinib-resistant” is preferable to be “imatinib-resistant”.

(11) In line of 91, “an indole based drug” is preferable to be “an indole-based drug”.

(12) In line of 135, is “one such a lead is one that” correct expression?

(13) In Figure 10, “FDA approved” is preferable to be “FDA-approved”.

(14) In line of 219, “Paclitaxel” is preferable to be “paclitaxel”, not at the beginning of a sentence. There are the same cases other than this.

(15) In line of 259, “doesn’t” is preferable to be “does not”.

That is all.

Author Response

Response to Reviewers' comments

We would like to thank the respected editors and reviewers for their comments on our work.

  • In the manuscript, the corrected sentences have been highlighted in blue color.
  • Improved manuscript editing and updating references has been carried out.
  • In this file, we have replied to all reviewers comments.

Comments of Reviewer 2#

The manuscript described the recent anti-angiogenics drugs. Angiogenesis is a process essential for cancer growth. The authors showed the structure-activity relationship (SAR) among the existing anti angiogenics drugs. According to such SAR analysis, a molecular modelling was also carried out. Thus, these findings will be useful for the drug development in anti-angiogenics drugs. Therefore, the manuscript is not too excellent to be published. In other words, the manuscript is so excellent that it should be published.

Reply to reviewer’s comment: We would like to thank the respected reviewer for his comments and opinion on our work.

Comments of the reviewer:

  1. It is true that heterocyclic fragments play a crucial role in pharmacodynamic aspects, but they do not always do in pharmacokinetic aspects due to the membrane permeability, solubility, and so on, mainly based on hydrophilicity. Can the calculation in silico pick up pharmaceutical structures sufficient to satisfy both pharmacodynamic and pharmacokinetic aspects?

Reply to reviewer’s comment: We would like to thank the reviewer for his keen interest in our work. According to previous studies pharmacokinetic (PK) and/or pharmacodynamic (PD) modeling describes a mathematical relationship among exposure and pharmacodynamic effect and governs how much exposure is needed, and for how long, to exert the effect. The aim of PK/PD modeling is to utilize the model to predict the exposure, PD effects under various experimental and therapeutic conditions and to ultimately make the most of the chance of success of drug candidates. The early development of PK/PD models is assisting both the discovery process and also the translation of preclinical data into the clinic. PK/PD modeling in discovery will continue to receive increasing focus in the future and will be expected to make a significant impact on drug discovery and development. in silico calculations of pharmacokinetics are important in directing design toward compounds. For more information, kindly, you may check the following reference:

- Alavijeh, M. S.; Palmer, A. M. J. C. O. I. D., The pivotal role of drug metabolism and pharmacokinetics in the discovery and development of new medicines. 2004, 5, 755-763.

  1. Which administration routes are suitable for anti-angiogenesis drugs? Is that Intravenous route or/and oral route?

Reply to reviewer’s comment: We would like to thank the reviewer for his question. Both routes are used in anti-angiogenics. For example, both sunitinib, axitinib, thalidomide and Lenvatinib are FDA approved oral drugs while other drugs such as Bevacizumab can only be administered by IV.

  1. Do intravenously administered angiogenesis inhibitors through VEGFR on the endothelial surface elicit their activity without being transported across the membrane? In that case, a molecular modelling might be conducted just based on pharmacodynamic aspects.

Reply to reviewer’s comment: We would like to thank the reviewer for his question and insight. While the VEGFRs act on the endothelial surface, their pharmacokinetics play a primary role in their activity as a large number of the FDA approved VEGFRs are orally administered.

  1. According to the molecular modelling results, do the existing anti-angiogenics drugs share the same binding sites, in each category such as VEGFR inhibitors and MMP inhibitors?

Reply to reviewer’s comment: Few drugs in each category act on same binding site but the majority act on different binding sites. This leads to the need for combination therapy in cancer patients in order to stop the progression of angiogenesis. For more information, kindly, you may check the following references:

-Rajabi, M.; Mousa, S. A. J. B., The role of angiogenesis in cancer treatment. 2017, 5, (2), 34.

- Grothey, A.; Galanis, E. J. N. r. C. o., Targeting angiogenesis: progress with anti-VEGF treatment with large molecules. 2009, 6, (9), 507.

  1. The quinolin-2-one analogues were introduced in this review. They were reviewed in the article titled as “The structural use of carbostyril in physiologically active substances” (https://doi.org/10.1016/j.bmcl.2015.06.027). Why don’t you add this in the reference list?

Reply to reviewer’s comment: Based on the reviewer’s suggestion, we have added the above-mentioned review to our references.

The following changes have already been done based on the reviewer’s suggestion

  1. In line of 21, “the structure activity relationship” is preferable to be “the structure-activity relationship”.
  2. In line of 60, “Molecular modeling” is preferable to be “molecular modeling”.
  3. In lines of 69-70, “VEGFR receptor” is preferable to be “VEGFR”.
  4. In line of 84, “Several Nitrogen based heterocyclic drugs” is preferable to be “Several nitrogen based heterocyclic drugs”.
  5. In line of 86, “Imatinib-resistant” is preferable to be “imatinib-resistant”.
  6. In line of 91, “an indole based drug” is preferable to be “an indole-based drug”
  7. In line of 135, is “one such a lead is one that” correct expression? corrected
  8. In Figure 10, “FDA approved” is preferable to be “FDA-approved”.
  9. In line of 219, “Paclitaxel” is preferable to be “paclitaxel”, not at the beginning of a sentence. There are the same cases other than this.
  10. In line of 259, “doesn’t” is preferable to be “does not”.

All modifications have been based on the reviewer’s suggestions. In addition, we have checked the whole manuscript.

Finally, we would like to express our thanks to the reviewers and editors for their efforts and comments which helped us to enhance the quality of our manuscript. We hope that our revised manuscript would be satisfactory and would be accepted for publication in Molecules.
